# Object Recognition and Grasping for Collaborative Robots Based on Vision

**DOI:** 10.3390/s24010195

**Published:** 2023-12-28

**Authors:** Ruohuai Sun, Chengdong Wu, Xue Zhao, Bin Zhao, Yang Jiang

**Affiliations:** 1College of Information Science and Engineering, Northeastern University, Shenyang 110819, China; sunruohuai@stumail.neu.edu.cn (R.S.);; 2SIASUN Robot & Automation Co., Ltd., Shenyang 110168, China; 3Faculty of Robot Science and Engineering, Northeastern University, Shenyang 110169, China; 4Daniel L. Goodwin College of Business, Benedict University, Chicago, IL 60601, USA; 20203734@stu.hebmu.edu.cn

**Keywords:** deep learning, collaborative robots, parallel networks, grasping detection

## Abstract

This study introduces a parallel YOLO–GG deep learning network for collaborative robot target recognition and grasping to enhance the efficiency and precision of visual classification and grasping for collaborative robots. First, the paper outlines the target classification and detection task, the grasping system of the robotic arm, and the dataset preprocessing method. The real-time recognition and grasping network can identify a diverse spectrum of unidentified objects and determine the target type and appropriate capture box. Secondly, we propose a parallel YOLO–GG deep vision network based on YOLO and GG-CNN. Thirdly, the YOLOv3 network, pre-trained with the COCO dataset, identifies the object category and position, while the GG-CNN network, trained using the Cornell Grasping dataset, predicts the grasping pose and scale. This study presents the processes for generating a target’s grasping frame and recognition type using GG-CNN and YOLO networks, respectively. This completes the investigation of parallel networks for target recognition and grasping in collaborative robots. Finally, the experimental results are evaluated on the self-constructed NEU-COCO dataset for target recognition and positional grasping. The speed of detection has improved by 14.1%, with an accuracy of 94%. This accuracy is 4.0% greater than that of YOLOv3. Experimental proof was obtained through a robot grasping actual objects.

## 1. Introduction

In recent years, the deployment of collaborative robots on a large scale in intelligent factories has increased with the convenience of human–robot collaboration to replace manual labor in undertaking 3C (Computer, Communication, and Consumer Electronics) manufacturing and intelligent handling. As a result, the intelligence, speed, and reliability of these robots have become core factors that impact the productivity and production quality of intelligent factories. Objectively sorting objects is a vital and habitual responsibility in intelligent production lines featuring collaborative robots. However, the conventional visual grasping technique solely operates on recognized, regularly formed pieces on the plane. This inability to accommodate intelligent factory products’ diverse and custom grasping needs is a significant limitation. Furthermore, manual sorting and grasping carried out by collaborative robots are prone to errors and omissions in intelligent production lines that involve many tasks and fast-paced operations. Such errors degrade the quality of the products in the intelligent production line and lead to confusion in the scheduling of upstream and downstream operations, ultimately leading to increased production costs. The vision-based collaborative robot grasping problem has become a widespread research topic in industry and academia.

To address these issues, implementing deep vision technology is a viable technical solution [1]. Deep vision technology accurately identifies and locates target objects on the intelligent production line. Simultaneously, it predicts the width and angle of the grasped objects, and the combination can accurately and efficiently complete the visual grasping task. Existing visual grasping techniques can accurately recognize and grasp objects with known types, regular shapes, and placement on a flat surface [2]. However, there rarely exist mature methods to identify object categories and simultaneously predict grasping scales and postures. Objective evaluations of grasping techniques are necessary to advance robotic grasping capabilities [3,4,5]. In this paper, we propose a visual grasping approach that utilizes the pre-trained YOLOv3 network and further training of the network using a self-built dataset to enhance generalization capability in object recognition for grasping. Additionally, the GG-CNN network was further trained using a self-built dataset on the pre-trained GG-CNN network to enhance its prediction accuracy for the grasping scales and attitudes of the target objects. By utilizing a combination of two networks, this method is capable of improved recognition of the object category to be grasped and selection of appropriate picking scales and postures. Its accuracy has been confirmed through a self-constructed dataset, and its effectiveness in a physical environment has been proven through collaborative robot experiments.

The main contributions of this paper are as follows:(a)This research proposes a parallel YOLO–GG deep vision network based on YOLO and GG-CNN.(b)This research utilizes data enhancement techniques to expand the dataset to increase the size of the dataset.(c)This article studies both the classification and grasping pose estimation issues in visual grasping.(d)This article conducts robot experiments to verify the proposed method in a real environment.

## 2. Related Works

In this section, we present research related to robot target recognition and grasping pose estimation, alongside advanced deep learning algorithms.

In intelligent production lines, the sorting of objects is a composite task comprising of two subtasks. First, targets must be classified based on their categories, which requires different sorting strategies such as varying speeds and destinations for each category. Then, after determining the category of the target, we grab and move the target to a specific location. We prioritize the orientation of the object and the opening width of the clamp in this process.

### 2.1. Target Classification and Detection

The objective of target detection is to identify the class and location of a target in a specified scene. Deep learning has yielded promising results in this area. The progress in computing capability combined with advances in deep learning has led to many fruitful outcomes in the field of target detection [6]. In the domain of 2D target detection, two categories of algorithmic approaches exist: the two-stage method exemplified by R-CNN and Faster R-CNN and the single-stage method represented by YOLO and SSD. Each type of approach has different priorities; the two-stage method focuses on accuracy but exhibits slower detection speed, whereas the single-stage method prioritizes detection speed over a certain degree of detection accuracy reduction.

In recent years, researchers have focused more on one-stage methods than two-stage methods because of their more concise designs and competitive performance [7]. Based on R-CNN and YOLO, Lin introduced RetinaNet, which concentrates the detector on hard-to-classify targets during training using the Focal Loss function. This leads to a detection accuracy that is comparable to Faster R-CNN. Meanwhile, Li proposed RepPoints, which employs a point set to predict targets and introduces spatial constraints for penalizing outliers during adaptive learning; the algorithm outperforms Faster R-CNN. Law and Deng departed from the previous assumption and considered target detection as the issue of key points. They proposed CornerNet and achieved excellent outcomes during that time. Zhou proposed CenterNet, which utilizes anchors only at the current target location rather than the entire image. Additionally, CenterNet does not require NMS for further filtering and achieves higher accuracy compared to previous methods. Faster R-CNN is a prevalent region-based, two-stage target detection method in computer vision. Its previous versions include R-CNN and Fast R-CNN. The basic idea of Fast R-CNN is to use a continuous convolutional layer to derive a feature map from the input image. A selective search algorithm is then applied to obtain region proposals, followed by pooling to adjust the proposed regions to a fixed size. Finally, the regions are inputted to the fully connected layer for classification and regression.

### 2.2. Target Grasping Pose Estimation

Obtaining visual information from various sensors is a more convenient alternative to obtaining a precise 3D model of an object. The research literature demonstrates excellent results in target classification research [8,9,10] and in target detection research [11,12], where the ability of deep learning to automatically learn image features has been extensively investigated. Many researchers have conducted extensive research in the area of robot grasping detection. Y. Domae proposed a technique to acquire the best grasping pose by developing a model of both the gripper and the target to be grasped, using a depth map. Suitable for settings with varying placement of multiple targets, the gripper model utilizes two mask images [13,14,15]. One of the images describes the contact region where the target object should be placed for stable grasping. The second image describes the collision region that should not be occupied by any other objects during the grasping process, in order to avoid collisions. The measure of graspability is calculated by convolving the mask image with a depth map that has been converted into a binary form. The threshold for each region differs based on the minimum height of 3D points in the region and the length of the gripper [16,17,18]. The proposed approach is appropriate for general objects since it does not presume any 3D model of the object [19,20]. Jeffrey Mahler developed Dex-Net2.0, which employs probabilistic models to create a comprehensive point cloud and robust grasping plans with precise labeling. Dex-Net 2.0 designs a deep grasp quality convolutional neural network (GQ-CNN) model and trains it to evaluate the robustness of grasping with candidate grasp planning and point cloud. The GQ-CNN model allows for obtaining candidate grasping plans by using edge detection on the input point cloud. By sampling these candidate plans, the most robust grasping can be performed based on the GQ-CNN estimation, enabling the planning of grasping on the actual robot. However, these efforts need two or more steps despite being capable of predicting object poses in cluttered scenes.

### 2.3. Related Paper Work

Grasping tasks that take place on an intelligent factory production line fall into the category of planar grasping. This scenario involves a robotic arm positioned vertically downward and grasping the target object from a single angle, such as when sorting and palletizing in an industrial setup. In 2D planar grasping, the target object is located on the planar workspace, and grasping is constrained by one direction, i.e., the support plane constraint. To address the inadequacies of the algorithms above, this paper introduces a parallel deep vision network utilizing YOLO and GG-CNN. In this scenario, the height of the gripper remains constant, and its orientation is perpendicular to a plane. Consequently, assessing gripper contact points and quadrangles with orientation is essential. This method allows end-to-end simultaneous detection of object-bounding boxes and the estimation of grasping locations using hierarchical features. By employing this strategy, even complex images can be processed through the deep neural network just once, allowing for optimal grasping of each target object in unstructured and cluttered environments.

The sections of this paper are organized as follows: Section 2 introduces the development of visual grasping based on deep vision. Through deep vision technology, this paper aims to build an automated picking system that can monitor categories of the objects on the production line in real time and give the appropriate attitude and opening width to grasp the objects. Section 3 introduces a composite network based on YOLOv3 and GG-CNN, which decomposes the visual grasping task into two sub-tasks of target detection and grasping pose estimation, which are handed over to the two networks for parallel processing. The generalization capability of the objects to be grasped is achieved by additional training on self-built datasets to the pre-trained model through a migration learning approach. Section 4 presents the experimental results of the proposed algorithm on the self-built dataset and the collaborative robot. Section 5 analyzes the final results of the proposed method. Based on the idea of “specialization in a specialized field”, this paper aims to decompose the complex visual grasping task into independent subtasks and to take advantage of the strengths of each sub-network through the parallel network structure. The decomposition and parallel execution of tasks can improve the efficiency of task execution in industrial application scenarios. This technology has a specific promotion value in the industrial application environment of visual grasping, which helps to meet the diversified and customized demands for visual grasping in intelligent factories.

## 3. Overview of Robotic Grasping

### 3.1. Grasping System

Figure 1 shows the deep vision grasping system used in this article. It consists of two primary modules: the target grasping information detection module and the robot grasping control module.

(a)The target grasping information detection module consists of an RGB camera and a topside computer, and both of them are connected via USB. When the end of the robot moves over the object to be grasped, the topside computer issues an instruction and processes the generated image. As shown in Figure 1, the camera is fixed on the end of the robot. The orientation of the shot is usually perpendicular to the plane of the object.

**Figure 1 sensors-24-00195-f001:**
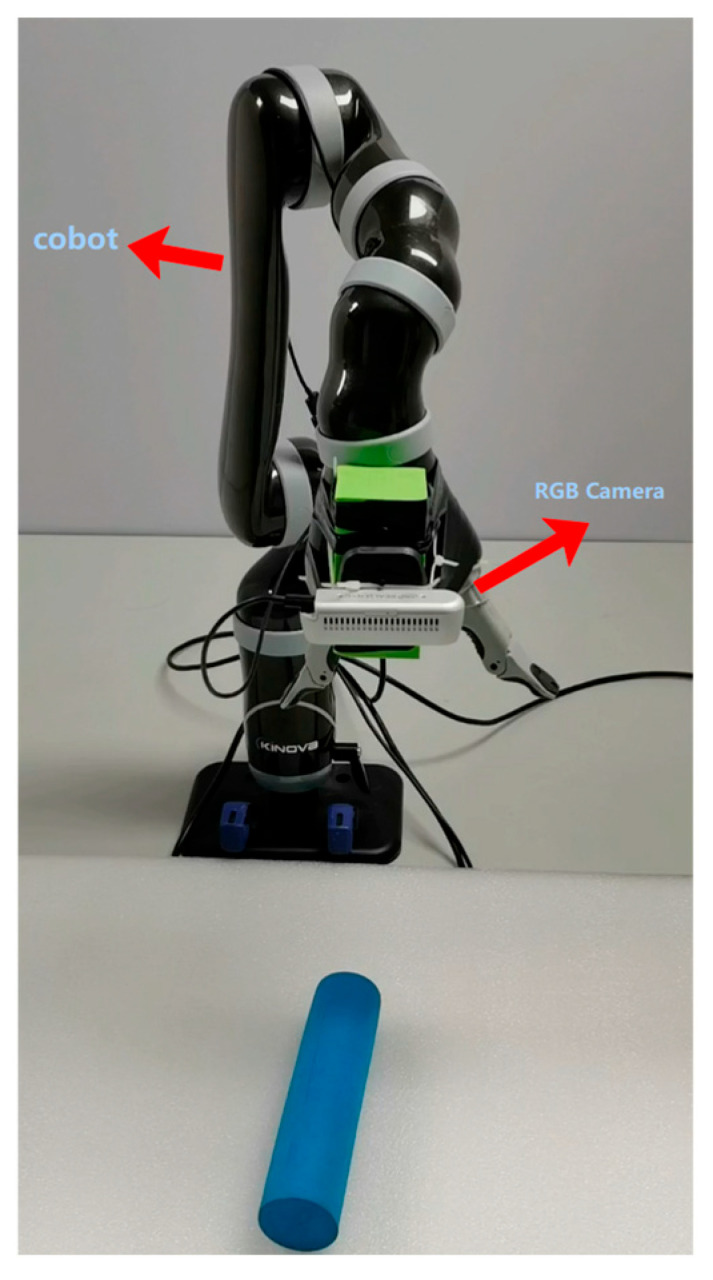
The relative location of the robot and the RGB camera in the workspace.

The target grasping information detection module retrieves RGB and depth images of the scene through the RGB-D camera. These images are pre-processed to match the input format of the deep learning network. Afterward, the deep learning inference prediction network generates images for mass, angle, and width, which are used to determine the optimal placement of the objects.

(b)The robot grasping control module utilizes information generated by the target grasping information detection module, which communicates the required actions to the robot by using trajectory planning and the ROS interface of the controller. The network model recognizes grasping information for the purpose of controlling the robot and performing a real-time plan and adjustment of the position and attitude of the grasping target.

### 3.2. Datasets and Preprocessing

This paper utilizes the Cornell Grasping and COCO datasets to create the NEU-COCO datasets for target recognition and position grasping. However, due to factors such as the limited number and type of samples, the dataset needs to catch up to the large-scale dataset standards, leading to model underfitting and ineffective feature learning by the network. To overcome this issue, we utilize data enhancement techniques. Before training, applying the same enhancement techniques to the dataset is necessary.

Data enhancement includes sample scaling, wherein the standard dataset size of 640 × 480 is reduced to 320 × 320. Additionally, there is a sample translation to adjust the position of target objects in conformity with the environment. Finally, sample rotation is used to rotate the sample, which enhances the robustness of the network.

It is necessary to consider that the multi-target field of view in the image is quite large. The object might exceed the boundary in cases of rotation and translation, resulting in an invalid capture box. To prevent this, the image should be resized and padded to ensure the object does not deform and remains in the field of view. Invalid capture boxes should be eliminated after enhancement. Additionally, each sample must contain a capture box and object category information, and the default training difficulty of all samples should be the same. Every sample includes the capture box, information on the object category, and training difficulty, which is the same for all samples by default.

## 4. Visual Grasping Based on Parallel Neural Network Architecture

### 4.1. Target Detection Strategy and Network

In the sorting process of the intelligent factory, it is necessary to determine the category to establish the final storage location of the object post-grasping. It is important to verify whether the center position of the object outputted by the target detection network matches the grasping position outputted by the estimation network for the identified category. The accuracy of the object localization results and the success of the grasping process depend on the close match between these two results. The grasping pose estimation network outputs the grasping position, which is then transformed to the target position in the world robot coordinate system, and the grasping operation is performed with the output of the grasping width and angle.

Intelligent factory production lines require real-time detection to ensure production efficiency. The YOLOv3 algorithm, as a one-stage target detection algorithm, differs from others in its operation speed. It divides the picture into multiple grids, generates an a priori frame based on the anchor mechanism, and then generates the detection frame in a single step, greatly enhancing the prediction speed of the algorithm. The fast predictive speed of the algorithm is well suited for real-time visual grasping requirements in collaborative robots.

The YOLOv3 core network architecture utilizes convolution with no pooling or fully connected layers. It downsamples the feature map size by half with a stride of two, resulting in a feature map size of 1/32 of the input after five downsamplings. In addition, the network also provides more a priori frames through three scales and incorporates multi-continuous feature map information to predict objects with different specifications.

### 4.2. Visual Capture Based on Parallel Network Architecture

In the preceding section, we divided the 2D planar vision grasping task into two subtasks: target detection and grasping bit position estimation. We could not process the two subtasks concurrently due to the characteristics of the single network and input constraints. We implemented the subtasks separately using a parallel network structure and consolidated the outcomes to produce grasping instructions for the target in the output segment of the network. Figure 2 depicts the structure of the parallel network.

As shown in Figure 3, the depth camera captures the RGB-D image, and the parallel network structure improves the processing speed by performing the target detection and pose estimation tasks simultaneously. The RGB image is fed into the YOLOv3 network for target detection. The feature map size is then reduced by 5-fold downsampling, resulting in a 1/32 of the input map. Finally, the network outputs the recognized object types and locations. The highest confidence level item types are chosen to determine the final target coordinates $\left(x,y\right)$ for the grasping task. Coordinates $\left(x,y\right)$ refer to the position of the object in the image. It is the coordinates in the image coordinate system, as shown in Figure 4.

The GG-CNN network processes the depth map as input and produces a “grasping map” for every pixel of the input map after undergoing 3 convolution and 3 deconvolution layers. The “grasping map” includes the grasping quality (confidence) $Q_{\theta }$, the grasping width $W_{\theta }$, and the grasping angle $\Phi_{\theta }$. By combining the outputs, the grasping gesture estimation network can provide the final target position information $\left(x,y,\theta ,w,Q\right)$ of the gripper.

## 5. Experimental Results and Analysis

### 5.1. Network Evaluation Indicators

This paper addresses the issues of target detection, classification, and gesture in manipulators. The experimental evaluation focuses on these two aspects of classification and grasping.

(a)Classification evaluation.

This article utilizes common classification detection evaluation methods, namely precision (P) and mean average precision (mAP), to assess the effectiveness of the network proposed in this study. These evaluation metrics are defined as follows:(1)$$p=\frac{TP}{TP+FP}$$
(2)$$R=\frac{TP}{TP+FN}$$
(3)$$mAP=\int_{0}^{1}PdR$$
where TP (True Positive), FP (False Positive), and FN (False Negative) represent the number of grasping targets that are accurately localized, incorrectly localized, and unidentified, respectively. Therefore, (TP + FP) represents the total number of grasping targets detected by the network, and (TP + FN) represents the total number of grasping targets in the actual value. The mean average precision (mAP) is computed from the precision (P) and recall (R) curve, which reflects the relationship between detection accuracy and completeness at varying confidence thresholds.

(b)Grasping evaluation.

There are two types of evaluation metrics for this type of robotic grasping: point metrics and rectangle metrics. The point metric evaluates the success rate of grasping by measuring the distance between the center of the predicted grasping frame and the center of the labeled grasping frame. It employs a threshold method to determine whether a grasping frame is valid or not. If any of these distances is less than the specified threshold, the grasp is considered successful. However, this evaluation metric does not account for the angle of the grasping. The disparity between the forecasted angle of the capture box and the actual angle of the capture box is within 30 degrees. The Jaccard index between the anticipated grab frame $R_{p}$ and the authentic value $g_{t}R_{t}$ is over 25%, and the Jaccard index is calculated using the following equation:(4)$$J\left(R_{t},R_{p}\right)=\frac{\left|\left(R_{t}\cap R_{p}\right)\right|}{\left|\left(R_{t}\cup R_{p}\right)\right|}>0.25$$
where $R_{t}$ represents the ground truth grasping area, and $R_{p}$ represents the predicted grasping area. Following this definition, the Jaccard index is equal to the area of interest (IoU) threshold in target detection studies. This includes providing information about the center of the grasp and the width of the grasp spread. Since the true rectangle on the ground can define a broad range of rectangles that can be grasped, a predicted rectangle that overlaps the true rectangle on the ground by 25% is considered a good true rectangle on the ground if it is correctly oriented. Thus, our model is evaluated using the rectangle metric.

### 5.2. Target Classification and Grasping Experimental Analysis

In real-world scenarios related to target classification and grasping, this research conducted several experiments to choose 100 images to construct the test sets for predicting target recognition and grasping. The research comprised statistical analysis of both misdetections and omissions found in the predicted results of the network. Misdetections occur when the network incorrectly recognizes targets, while omissions refer to the failure to detect targets, also known as leakage. The experimental results are outlined in Table 1.

During the physical experiments, the YOLO–GG network demonstrated markedly superior performance in comparison to misdetection and omission. Additionally, during testing, the YOLO–GG network was able to achieve an inference speed of 35.7 frames, ultimately meeting the fundamental requirements for practical deployment. The AUROC is 0.945.

The hyperparameters used are listed in Table 2.

Subsequently, the experiments continued on the NEU-COCO dataset for target recognition and position grasping, and accuracy results were obtained as presented in Table 3.

Table 1 and Table 3 demonstrate that the method proposed in this article achieves a balance between speed and accuracy when compared to the classic method. The use of parallel networks results in improved performance in both classification and grasping subtasks.

After conducting multiple experiments, the YOLO–GG parallel deep learning network approach utilized in this paper proves to yield effective outcomes in field deployment. Furthermore, by testing the YOLO–GG parallel deep learning network on various datasets for different tasks, its generalizability is further confirmed.

### 5.3. Grasping Effect on Real Robot

A 6-DOF KINOVA robotic arm was selected as the robot body of this experiment. The upper computer system of the experiment adopts an i7 11700H CPU equipped with an Ubuntu16.04 operating system to achieve the motion control of the robotic arm through the ROS Kinetic application layer interface, and the camera adopts a RealSensed435i RGB-D camera.

In Figure 5, the grasping process of different objects is presented.

Figure 6 presents the impact of the suggested approach on the self-built NEU-COCO dataset for recognizing and positioning targets. The results indicate that targets were accurately classified, and mid-objects were the grasping positions, yielding more satisfactory outcomes.

Figure 7 shows the effect of grasping for physical objects using the method of this paper. During the grasping process, multiple objects are typically positioned close to each other. In the sorting workflow, it is necessary to categorize all captured objects and select the desired ones to be placed in the appropriate position.

The accuracy and speed of object recognition are related to the number of objects present in the camera’s field of view. We conducted 100 sets of experiments for two to four objects, respectively, and the statistics are shown in Table 4.

It can be seen that as the number of objects increases, both recognition accuracy and recognition speed decrease slightly. Additionally, the accuracy of grasping decreases. However, it maintains at a high level.

The practical value of the proposed method is evident in real-life application scenarios.

This article utilizes deep vision for object recognition and positioning. Considering that collaborative robots have strong interactivity with the environment, the perception of the environment may be a subject of our future research. Integrating depth vision and 3D point cloud information to improve the accuracy of object recognition and improve system stability is a research direction to be considered in the future.

## 6. Conclusions

With the widespread integration of collaborative robots in intelligent factories, the increasing diversity and customization of production tasks, such as smart handling and 3C manufacturing, has presented a challenge to the intelligence of robots. The conventional method of visual sorting combined with manual classification has limitations, such as a high occurrence of manual errors and low productivity, which fails to fulfill the requirements of modern intelligent factories. This paper examines a deep learning-based parallel network for recognizing and grasping targets to enhance the efficiency and precision of visual grasping by collaborative robots. The paper decomposes the visual grasping task into two subtasks: detecting the target and estimating the grasping pose. The YOLOv3 network processes the former, while the latter is processed by the GG-CNN network. The target recognition and grasping network can detect diverse unidentified objects and identify their categories and corresponding object bounding boxes. In order to fit the working environment of sorting in an intelligent factory, we conducted grasping experiments on multiple objects under working conditions. The accuracy of recognition and grasping was maintained at a high level; therefore, this method can be applied in an intelligent factory environment. The experimental results indicate that the suggested parallel network structure effectively manages visual grasping subtasks for various objects in a physical environment. The parallel network structure accelerates training. The effectiveness of grasping is also validated through actual robot grasping experiments.

## Figures and Tables

**Figure 2 sensors-24-00195-f002:**
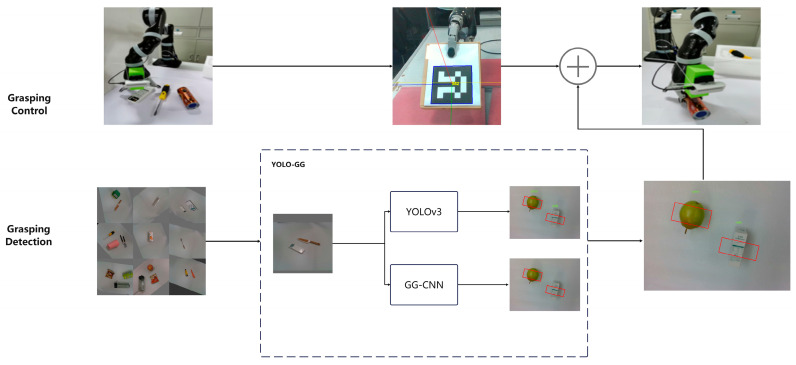
Grasping system.

**Figure 3 sensors-24-00195-f003:**
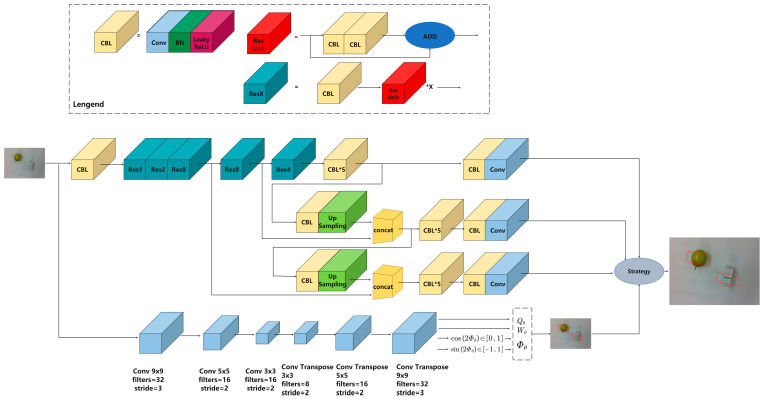
Parallel grasping network structure.

**Figure 4 sensors-24-00195-f004:**
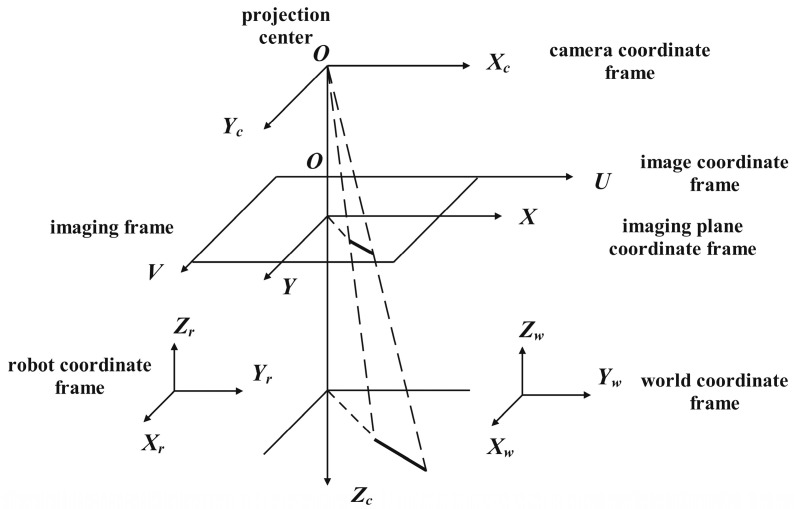
Schematic diagram of image coordinate system.

**Figure 5 sensors-24-00195-f005:**
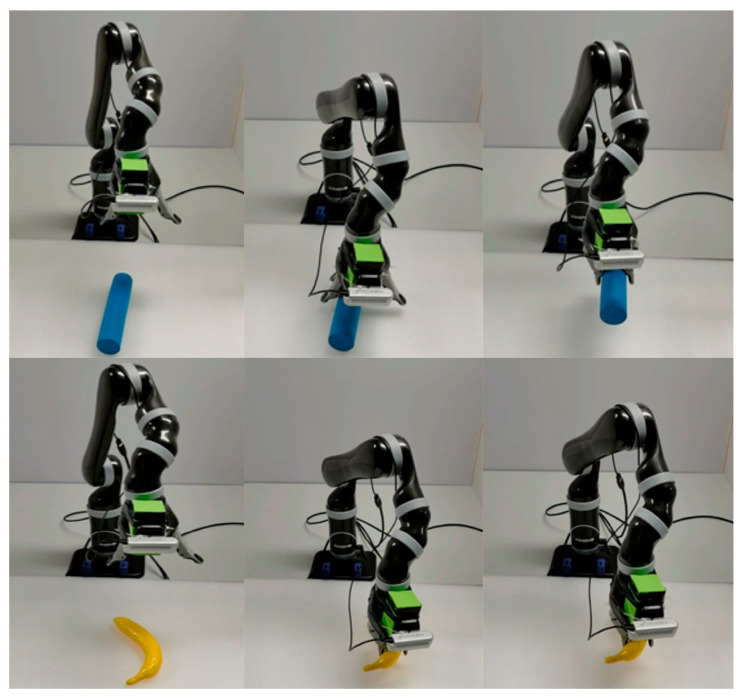
Grasping process of different objects.

**Figure 6 sensors-24-00195-f006:**
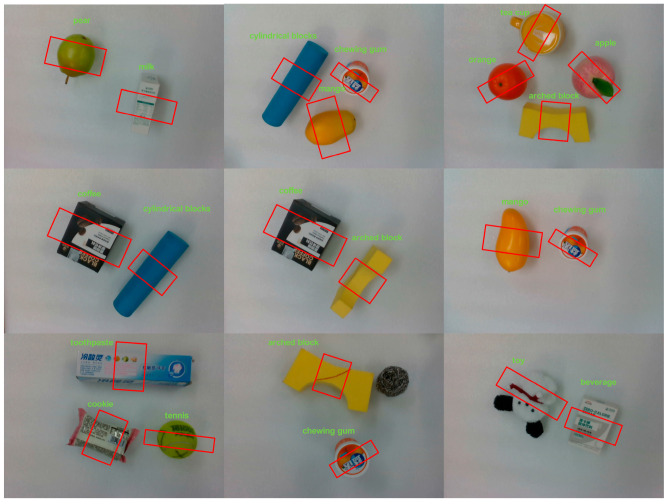
Target recognition results.

**Figure 7 sensors-24-00195-f007:**
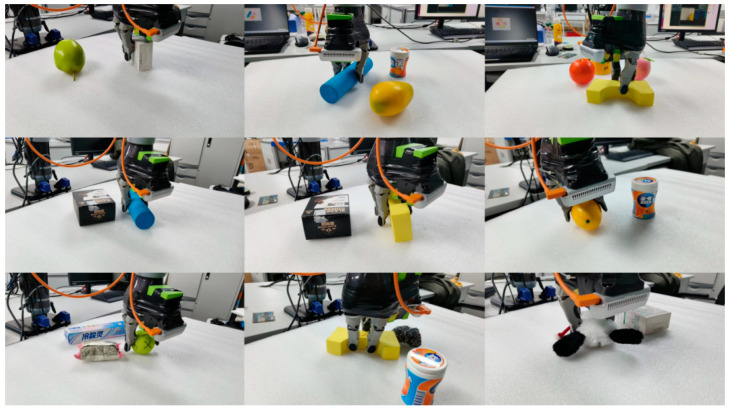
Grasping results.

**Table 1 sensors-24-00195-t001:** Experimental results of detection of false detections and missed detections.

Method	Self-Built Dataset	Number of Misdetections	Number of Omissions	Detection Time
YOLO–GG	NEU-COCO	2	4	28 ms
YOLOv3	NEU-COCO	4	6	32 ms
ConvMixer	NEU-COCO	12	13	29 ms
Resnet50	NEU-COCO	3	5	82 ms

**Table 2 sensors-24-00195-t002:** The hyperparameters used.

Learning Rate Decay	Epoch	Dropout	Batch Size	Optimizer
3, 0.33	150	0.2	6	SGD

**Table 3 sensors-24-00195-t003:** Target recognition and pose estimation on NEU-COCO dataset.

Method	mAP(%)
YOLO–GG	78.32
Faster R-CNN	71.10
YOLOv3	74.50
YOLOv5L	76.80
DCC-CenterNet	79.40
RDD-YOLO	81.10

**Table 4 sensors-24-00195-t004:** Statistics of the grasping results.

Number of Items	Recognition Accuracy (%)	Recognition Time (ms)	Capture Accuracy (%)
2	96	24	94
3	95	27	93
4	91	33	90

## Data Availability

Data are contained within the article.

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
