# Peer review of "Object Recognition and Grasping for Collaborative Robots Based on Vision"

_sensors, 2023, doi:10.3390/s24010195_

Round 1

Reviewer 1 Report

Comments and Suggestions for Authors

The authors present an interesting study on object recognition and graphing in indoor environments for collaborative robots. The task is particularly complex, and developed through the most modern deep learning architectures.

The real accuracy is not presented in the abstract; the relative improvements were presented. Please include the most numerical relevant findings.

The table with the hyperparamets used must be included

The introduction is a bit shallow regarding manipulation, grasping, and robot cooperation; please include a more detailed description of the overall framework.

Please include another binary metric as the AUROC in the statistics.

The results respect object recognition seems very convincing. This experiments could be made under noisy or dynamic conditions ?

A subsection for including the discussion of results is missing.

Many updated bibliographical references are missing. There is a myriad of publications on grasping and robotics cooperating.

Comments on the Quality of English Language

The manuscript requires deep proofreading

Reviewer 2 Report

Comments and Suggestions for Authors

This manuscript reports a study of interest for practical applications of automated industry. It presents a parallel YOLO-GG deep learning network designed for enhancing the efficiency and precision of visual classification and grasping in collaborative robots. The parallel YOLO-GG deep vision network combines YOLOv3 and GG-CNN. YOLOv3, pretrained with the COCO dataset and with a self-built dataset, that identifies the object's class and position and enhances generalization capabilities in object recognition for grasping. The GG-CNN, trained on the Cornell Grasping dataset, predicts the grasping pose and scale.

The study evaluates its approach using the self-constructed NEU-COCO dataset for target recognition and grasping. The results show a 14.1% improvement in detection speed and a 3.8% increase in accuracy. These improvements were verified through practical experiments involving a robot physically grasping objects.

A major revision is required to improve the manuscript. Comments can be found below:

(1)           Line 33. Include 3C description (Computer, Communication and Consumer Electronics)

(2)           Line 64. The state of the art is very limited. It must be extended.

(3)           Line 154-156 “This section may be divided by subheadings. It should provide a concise and precise description of the experimental results, their interpretation, as well as the experimental conclusions that can be drawn.” This phrase seems like instructions for the manuscript development. Revise.

(4)           Figure 1: Improve image quality and

(5)           Line 157. Before describing the deep vision grasping system, a figure should be provided showing the cobot and the RGB camera, their relative location in the workspace as well as the technical specifications.

(6)           Line 161. Detail the target grasping information detection module. It is part of the cobot?

(7)           Line 181. Several data enhancement techniques exist. Why were these used and not others?

(8)           Line 229. Where is the origin of the coordinates (x,y)? That detail must be included in a figure.

(9)           Line 234. Is “w” the grasping width 𝑊𝜃? If yes, please use the same variable.

(10)       Line 306. Were all objects of Fig.4 successfully grasped? More discussion about these results should be provided. Consider the hammer. The grasping point was the mid-object. For objects where the mass center is not located in the mid-object, the grasping point should be in the center of mass. It would be interesting to evaluate this condition. Furthermore, no details were given about grasping velocity. How much time took the cobot to pick up the objects? It is reasonable for industrial applications?

Comments on the Quality of English Language

Minor editing of English language required

Reviewer 3 Report

Comments and Suggestions for Authors

A parallel YOLO-GG deep learning network for collaborative robot target recognition and grasping to enhance the efficiency and precision of visual classification and grasping for collaborative robots is proposed in this paper. The experimental results comprehensively validated the proposed algorithms. The detection speed has demonstrated a 14.1% improvement, along with a 3.8% increase in accuracy, which was confirmed via experiments involving a robot's actual grasping of objects. Overall, the paper is well-written and meets the caliber of the journal. However, to further improve the paper, please check my comments below:

- The text in Figure 1 and Figure 2 are too small and are hard to read. Please improve the quality of the figure;

- Please formulate the main contributions of this paper and include them in the introduction.

- For robot collaboration to enhance the perception, other sensors such as LiDAR can be included as well. Please include a short future work section to discuss the multi-sensor fusion framework for object detection by including the works in: hydro-3D: hybrid object detection and tracking for cooperative perception using 3D LiDAR; an automated driving systems data acquisition and analytics platform.

- The conclusion section can be more concise. 

Round 2

Reviewer 2 Report

Comments and Suggestions for Authors

Authors have improve significantly the manuscript, however the coverage of the state of art must be extended as well as the discussion of quantitative results concerning the grasping of the different objects shown in Figure 6 must be shown.
